# Cylindrical Microparticles Composed of Mesoporous Silica Nanoparticles for the Targeted Delivery of a Small Molecule and a Macromolecular Drug to the Lungs: Exemplified with Curcumin and siRNA

**DOI:** 10.3390/pharmaceutics13060844

**Published:** 2021-06-07

**Authors:** Thorben Fischer, Inga Winter, Robert Drumm, Marc Schneider

**Affiliations:** 1Department of Pharmacy, Biopharmaceutics and Pharmaceutical Technology, Saarland University, Campus C4 1, 66123 Saarbruecken, Germany; thorben.fischer@uni-saarland.de (T.F.); s9wiing@stud.uni-saarland.de (I.W.); 2INM-Leibniz Institute for New Materials, Campus D2 2, 66123 Saarbruecken, Germany; robert.drumm@leibniz-inm.de

**Keywords:** nanotechnology, pulmonary delivery, TNFalpha inhibition, RNA release, inhalation therapy, aspheric drug delivery systems, curcumin delivery, synergistic effect

## Abstract

The transport of macromolecular drugs such as oligonucleotides into the lungs has become increasingly relevant in recent years due to their high potency. However, the chemical structure of this group of drugs poses a hurdle to their delivery, caused by the negative charge, membrane impermeability and instability. For example, siRNA to reduce tumour necrosis factor alpha (TNF-α) secretion to reduce inflammatory signals has been successfully delivered by inhalation. In order to increase the effect of the treatment, a co-transport of another anti-inflammatory ingredient was applied. Combining curcumin-loaded mesoporous silica nanoparticles in nanostructured cylindrical microparticles stabilized by the layer-by-layer technique using polyanionic siRNA against TNF-α was used for demonstration. This system showed aerodynamic properties suited for lung deposition (mass median aerodynamic diameter of 2.85 ± 0.44 µm). Furthermore, these inhalable carriers showed no acute in vitro toxicity tested in both alveolar epithelial cells and macrophages up to 48 h incubation. Ultimately, TNF-α release was significantly reduced by the particles, showing an improved activity co-delivering both drugs using such a drug-delivery system for specific inhibition of TNF-α in the lungs.

## 1. Introduction

According to the World Health Organization (WHO), chronic respiratory diseases are among the most widespread diseases worldwide with a high mortality rate, which is expected to increase further in the future [1]. With several million cases, cystic fibrosis, idiopathic pulmonary fibrosis, tuberculosis, chronic obstructive pulmonary disease (COPD) and asthma represent the most common diseases for this clinical pattern [2], leading to a permanent inflammation [3,4,5,6] and thus affect the lung function over time [6,7]. This proinflammatory status is caused by dysregulation of gene expressions, resulting in an imbalance of produced transcription factors [8], which translates into increased release of proinflammatory mediators such as cytokines [9]. The cell populations such as macrophages and immunologically active epithelial cells, which are mainly responsible for the production and release of these messenger substances, are located in the alveolar region of the lungs [10]. In a chronic inflammatory state, the continuous secretion of cytokines and proinflammatory mediators may lead to additional deterioration of the lung tissue. The interruption of the hyperinflammatory, ongoing release of cytokines will support treatment [11].

To reach the deep lungs, several inhalation systems have been developed to generate the highest possible efficiency. Most commonly used for pulmonary delivery are dry powder inhalers (DPIs), metered dose inhalers (MDIs), soft mist inhalers (SMIs) and nebulizers [12]. Having a closer look at DPIs, usually powder mixtures of the micronized drug particles are mixed with inert substances such as lactose, mannitol, trehalose, etc. in order to deliver the drug as efficiently as possible to its target [13]. Spray-drying is often applied, providing suitable particles for inhalation, although many commercial formulations are reported to have a lung deposition as low as 14% [14].

A relatively new drug carrier for lung application is mesoporous silica particles. Although a novel approach, they are already commercially offered as a pulmonary transport system due to their various positive characteristics [15]. Especially, mesoporous silica nanoparticles (mSNPs) are more biocompatible due to their structure, which results in faster degradation and thus a lower toxic potential in comparison to amorphous silica nanoparticles [16]. In addition, the mSNPs can be loaded in high quantities [17] with labile molecules, such as curcumin, as the pores create a protective environment [18]. This is also one of the reasons why such particulates were already intensively used as a drug delivery system for various application routes for diverse targets [19,20,21].

The filled pores have only a small area where the outer medium comes into contact with the drug, leading to a slower release [22]. In addition, to further modify the release and better protect labile drugs in the pores, the surface of the mSNPs can be coated with polymers. Here, the strong negative surface charge of the silica particles can be exploited [18]. An established method for this is the layer-by-layer technique [23]. This allows not only the surface to be loaded with drugs, but also polymers to be used as stabilizers for the formation of larger particle constructs [24,25]. This property can be used to manufacture drug delivery systems for pulmonary application, where an aerodynamic diameter of 1–5 µm must be achieved to reach the deep lungs [26]. 

To generate a stronger cell response and inhibit the release of proinflammatory cytokines, mSNPs can be loaded in the pores with multiple anti-inflammatory agents, such as the natural compound curcumin, and also a negatively charged oligonucleotide, such as a specific siRNA, on the surface to generate an improved effect [27]. However, since these two drugs have low stability and critical pharmacokinetic properties [28,29,30], targeted transport into immunologically active cells is of particular importance to prevent disintegration and also reduce side effects [31]. The use of aspheric particles for pulmonary delivery represents a possible mode of delivery for this purpose since the change in shape affects the interaction with immunological cells such as macrophages [32]. Here, a cylindrical shape has been established, since a change in the length-width ratio, also called aspect ratio, leads to a modified cell uptake into macrophages [33]. In addition, the specific surface area of cylindrical microparticles is significantly larger than the surface area of their spherical counterparts, which allows higher loading quantities. This drug delivery system has already shown that loading with a specific siRNA sequence against TNF-α is possible and a significant reduction in this cytokine release from macrophages could be achieved, but also showed the need for improvement [24].

The idea was to provide a drug carrier system able to interact with alveolar macrophages and allow an efficient reduction in TNF-α release. Delivery of sensible macromolecular drugs in combination with a supportive conventional small molecule is a novel approach. The formulation shall load and protect sensible drugs and deliver it via oral inhalation. The exploitation of the advantages of mSNPs assembled to inhalable cylinders for siRNA delivery is an approach not yet described. The aim was to develop a nanostructured cylindrical drug carrier system composed of mSNPs to ensure the directed transport of siRNA in macrophages. Supplemental curcumin as an additional anti-inflammatory agent was used to achieve a synergistic effect and further reduce the release of cytokines. In this work, we focused on the inhibition of tumour necrosis factor alpha (TNF-α) as one of the most prominent and active cytokines in the inflammation cascade [34]. Such a system should allow enhancement of the effect combining the two drug types and may underline the potential of such a carrier system for future applications.

## 2. Materials and Methods

### 2.1. Materials

Mesoporous silica nanoparticles with a pore size of 4 nm and a diameter of 200 nm (748161-1G) as well as curcumin (C1386-50G), branched polyethyleneimine (bPEI) 25 kDa (408727-100ML), L-leucine (L8000-25G) and 3-aminopropyltrimethoxy silane (APTS) (440140-100ML) were purchased from Sigma Aldrich (Steinheim, Germany). Dextran sulphate 10 HS (DB008) was purchased from TdB Labs (Uppsala, Sweden). Nuclepore^®^ Track-Etched Membranes with a pore size from 0.1 µm (WHA110605) and 3 µm (WHA110612) in diameter were purchased from Whatman plc (Kent, UK). RPMI-1640 cell culture medium (R8758-500ML), Hanks’ Balanced Salt solution (HBSS) (H9394-500ML), dimethyl sulphoxide (DMSO) (472301-500ML), phorbol 12-myristate 13-acetate (PMA) (79346-5MG), lipopolysaccharides from Escherichia coli (LPS) (L2880-10MG), 3-(4,5-dimethylthiazol-2-yl)-2,5-diphenyltetrazolium bromide (MTT) (M5655-100MG), foetal bovine serum (FCS) (F2442-500ML) and ethidium bromide solution (E1510-10ML) were obtained from Sigma Aldrich Life Science GmbH (Seelze, Germany). Nuclease-Free Water (AM9939), TNF alpha Human ELISA Kit (KHC3011) and tetrahydrofuran (109-99-9) were obtained from Thermo Fisher Scientific Inc. (Darmstadt, Germany). Silencer^®^ Pre-designed siRNA (sequences: Sense (5′-GGACGAACAUCCAACCUUCtt-3′) Antisense (5′-GAAGGUUGGAUGUUCGUCCtc-3′)) (AM16706) was purchased from Ambion Inc (Austin, TX, USA). Silencer™ Select Negative Control No. 1 siRNA (Sequences: Sense (5′-UAACGACGCGACGACGUAAtt-3′) Antisense (5′-UUACGUCGUCGCGUCGUUAtt-3′)) used as negative control for siRNA (scrambled siRNA) (4390844) was purchased from Invitrogen™ (Carlsbad, CA, USA). 

### 2.2. Rhodamine B Functionalization of mSNP for Visualization

In order to improve the detection and quantification of the colourless nanoparticles in various experiments, a fluorescent dye was added. Therefore, 3-aminopropyltrimethoxy silane (APTS) was covalently bound to the hydroxy groups on the surface of the silica particles [35]. Beforehand, the primary amine group of APTS was bound with the carboxy group of rhodamine B to form an amide. This compound was then applied to functionalize the surface of the nanoparticles by covalently attaching the rhodamine B silane to the surface via the methoxy silanes. For this, a 3% rhodamine B solution in chloroform was prepared and mixed with APTS. This mixture was heated for 30 min to 60 °C under stirring. The remaining chloroform was then evaporated, and the residue was dissolved in ethanol and adjusted to pH 3.5 using 0.5 N hydrochloric acid. The resulting solution was then allowed to react for 1 h while stirring. Subsequently, mSNPs were mixed in a ratio of 1:10 with the prepared solution and incubated for 6 h while shaking in a thermal mixer at 60 °C. The suspension was then washed twice with ethanol until no unreacted fraction of rhodamine B was present [36]. This should result in fluorescent, violet-stained nanoparticles with an excitation wavelength of 545 nm and an emission wavelength of 567 nm, allowing straightforward quantification.

### 2.3. BET Measurement to Evaluate Pore Volume and Loading with Curcumin

The measurement of the adsorption isotherm after Brunauer, Emmett and Teller (BET) determines the monolayer concentration of an adsorbed gas on the sample surface. From this value, the specific surface of the sample can be determined. The physical adsorption observed is based on van der Waals interactions between the adsorbed gas molecules and the adsorbing sample surface. Usually, nitrogen is used as adsorption gas in this method, as far as the temperatures and the sample allow it. The specific surface area determined in a BET measurement also includes the surface area of pores in porous materials and is therefore in relation to the relative surface area. Gas adsorption thus enables the determination of the size and volume distribution of micropores. In addition, comparative measurements can be used to directly check loading tests and to determine the loading quantities. For sample preparation, the powder was first dried in vacuum at an elevated temperature followed by the measurement at the boiling point of nitrogen. After recording at least three data points, the BET value can be determined using the BET adsorption isothermal equation (Equation (1)). The amount of adsorbed gas is correlated with the total surface area of the particles, including pores in the outer surface [37].
(1)$$\frac{p}{V_{a}\cdot \left(p_{0}-p\right)}=\frac{1}{V_{m}\cdot C}+\frac{C-1}{V_{m}\cdot C}\cdot \frac{p}{p_{0}}$$p = partial vapor pressure of the adsorbed gas in equilibrium with the surface at 77.4 K (boiling point of liquid nitrogen) (Pa);p_0_ = saturation pressure of the adsorbed gas (Pa);V_a_ = volume of adsorbed gas under standard conditions (273.15 K and 1.013 × 10^5^ Pa) (mL);V_m_ = volume of the adsorbed gas, to enable monolayer formation on the sample surface (mL);C = dimensionless constant related to the adsorption enthalpy of the adsorbing gas on the powder sample.


For sample preparation, 0.2951 g of mSNPs were added to a quartz cell and evacuated for 10 h at 130 K (degassing) to remove physically adsorbed water and volatiles on the surface of the particles followed by the adsorption cycle at 77.4 K. As reference, the quartz cell without sample was degassed under the same conditions and subsequently an adsorption cycle was performed. Nitrogen was used as adsorbate gas as it is inert and interacts with most solids. The pressure difference between the sample and the standard correlates with the amount of adsorbed nitrogen. From this, the surface area and the pore volume can be determined.

### 2.4. mSNP Pore Loading with Curcumin

To load the pores of the mSNPs, 2 mg of curcumin was dissolved in 1 mL of acetonitrile and subsequently 10 mg of nanoparticles were added to the solution. This suspension was first homogenized in an ultrasonic bath and then heated for 1 h at 90 °C and 500 rpm to evaporate the solvent. By reducing the volume of the solvent, the hydrophobic drug is forced into the pores of the nanoparticles, due to the increasing curcumin concentration by solvent evaporation. Only a small amount of the curcumin attaches to the hydrophilic surface of the nanoparticles and was removed by washing with water for three times. To confirm this hypothesis, this loading method was first performed with non-porous, non-fluorescently labelled silica particles. 

### 2.5. Determination of Colloidal Properties

Zetasizer Nano ZS (Malvern Panalytical, Malvern, UK) was used to determine hydrodynamic diameter via dynamic light scattering. Here, the polydispersity index (PDI) was also determined describing the size distribution of the nanoparticles. The same device was used to determine the zeta potential according to the principle of laser doppler micro electrophoresis. For sample preparation, a stock solution of 1 mg/mL was diluted 1:20 with MiliQ^®^ water and measured three times, with each of the three measurements including 12 runs.

### 2.6. Preparation of Cylindrical Microparticles Composed of mSNPs

To produce the nanostructured cylindrical microparticles (microrods), the mSNPs must first be assembled in a cylindrical shape. Therefore, a template-assisted approach was used [25,38] to transfer the nanoparticles into a template membrane with a defined length and width of 10 µm and 3 µm. This shaping membrane was placed into a filter holder system, which was sealed with a silicon ring. A second membrane was added as a blocking membrane underneath the template membrane. Thus, the mSNPs remained in the pores of the template membrane, allowing only the water of the suspension to pass through the small pores (d = 0.05 µm) of the blocking membrane. A syringe containing 500 µL of a 1.5 mg/mL mSNP suspension was connected to the filter holder, continuously injecting particles into the pores of the template membrane with a flowrate of 100 µL/min created by a syringe pump (Legato 210; KD Scientific, Holliston, MA, USA). This process was repeated three times to achieve uniform filling. Afterwards, the template membrane was washed with a lint-free tissue to remove the particles on the top and bottom of the membrane.

Since the particles are only loosely assembled and not connected strongly to each other in the pores, the layer-by-layer technique was used to increase the ionic interactions and thus stabilize the microrods [39,40]. Polycationic substances branched polyethyleneimine (bPEI) was used and dextran sulphate was applied due to its negative charge. These substances were used because they offer relevant properties. Dextran sulphate is highly branched and thus yields many anionic groups due to the high number of sulphate groups per glycosyl unit (approximately 2.3) [41] and it is biocompatible [42]. bPEI was used because this polymer has a strong buffering capacity, which is described to promote osmotic swelling of the lysosome and thus destabilizes the membrane of the vesicle. Through this so-called “’proton sponge effect” the diffusion of the active ingredients into the cytosol shall be facilitated [43] allowing reaching the target in the cytosol [34]. In a first step, the template membrane with the infiltrated mSNP was added to 2% bPEI solution for 12 min to allow the cationic polymer to adhere to the negatively charged surface of the nanoparticles. After a subsequent washing step with water, the membrane was placed in a 2% dextran sulphate solution for 12 min. After a second washing step, this cycle of layering was repeated. A total of three double layers (PEI and dextran sulphate form one double layer) of the polymers was necessary to generate a stable formulation. To load the particles with siRNA, the dextran sulphate solution was replaced by an 0.05% siRNA solution in the last layering step. To release the stabilized formulation from the polycarbonate template membrane, it was dissolved in tetrahydrofuran (THF). The THF insoluble microrods were then centrifuged at 6500× *g* for 7 min and the supernatant containing dissolved polycarbonate was discarded. Washing with THF was performed five times in total and after the last step the inorganic solvent was evaporated, resulting in a dry powder. Since the oligonucleotide used is sensitive to nucleases, the entire production process was performed under aseptic conditions.

### 2.7. Aerodynamic Characteristics Analysed by Next Generation Impactor

Since the developed formulation was intended for inhalation into the lungs, the aerodynamic properties were determined. Therefore, a Next Generation Impactor (NGI) (Copley Scientific, Nottingham, UK) with standardized settings was used [44]. An M1A flowmeter (Copley Scientific, Nottingham, UK) was used to adjust a continuous airflow of 60 L/min ensuring aerosolization of the microparticles [45,46]. These were coated with L-leucine at the surface [47] before application to reduce hygroscopicity [48] and increase surface roughness to improve flight characteristics [49,50]. Approximately 3 mg of the coated formulation was then filled into a capsule, placed in a HandiHaler^®^ (Boehringer Ingelheim, Ingelheim, Germany) and punctured. An air flow of 60 L/min was then generated for 4 s by a vacuum pump (Erweka, Langen, Germany) to aerosolize the formulation into the NGI. The particle concentrations in each stage were then analysed using a microplate spectrophotometer (TecanReader^®^ infinite M200, Tecan, Männedorf; Switzerland) to determine the Mass Median Aerodynamic Diameter (MMAD), Fine Particle Fraction (FPF) and Geometric Standard Deviation (GSD). The calculation of these values was performed analogous to Abdelrahim et al. [51] as already described in different publications [52,53].

### 2.8. Differentiation of Human Monocytes (THP-1) into Macrophage-Like Cells

To determine the interaction between the developed formulation and immunologically active cells, the human monocytic cell line THP-1 was used. In a first step, this cell line must be differentiated into M0 macrophage-like cells. Therefore, the cells were incubated with a mixture of RPMI 1640 medium with an addition of 10% foetal calf serum (FCS) and 50 ng/mL phorbol 12-myristate 13-acetate (PMA) for three days in a humidified atmosphere of 5% carbon dioxide at 37 °C. Afterwards, the cells were washed twice with HBSS and supplemented with medium without PMA. The cells resulting from the differentiation process represent a M0 macrophage-like cell line (dTHP-1) [54]. As the method is well established, the successful differentiation was only monitored by light microscopy evaluation of the form of the cells and the fact that they turn into adhesive cells growing on the support. 

### 2.9. Human Alveolar Basal Epithelial Cell Line A549

Since not only immunologically active cell types interact with the formulation during pulmonary application of drug delivery systems, the A549 cell line was used as model for pulmonary epithelial cells to analyse possible reactions [55]. To cultivate the cells, they were incubated with RPMI 1640 medium with an addition of 10% FCS in a humidified atmosphere of 5% carbon dioxide at 37 °C.

### 2.10. Determination of Cell Viability by MTT Assay

To get a first impression of how the cells interact with the developed formulation, a cytotoxicity test was performed. For this study, the 3-(4,5-Dimethylthiazol-2-yl)-2,5-diphenyltetrazoliumbromid (MTT) assay was applied to colourimetrically determine the metabolic activity of the cells after incubation with the drug delivery system [56]. For this, the cell lines A549 and dTHP-1 described above were cultivated in 96-well plates with a concentration of 2 × 10^4^ cells per well. Subsequently, the cells were incubated for 4, 24 and 48 h with a particle concentration of 10, 50 and 100 µg/mL, respectively. To obtain these concentrations, the formulation was redispersed and diluted in RPMI 1640 + 10% FCS. After different periods of time, the supernatant was removed and replaced by MTT reagent incubating for 4 h. In a final step, the supernatant was removed and replaced with DMSO to dissolve the resulting formazan crystals. Cell viability was then determined by measuring the absorbance at 550 nm after 20 min using microplate spectrophotometer. Untreated cells grown in medium and Triton-X treated cells were used a negative and positive control, respectively. 

### 2.11. In Vitro Release of siRNA and Curcumin under Phagolysosomal Conditions

When incubating cells with particles, usually not 100% of the active ingredient is released in time [57]. Therefore, it is unclear what amount is available for drug action. Consequently, a release study in phagolysosomal simulant fluid (PSF) [58] was performed to determine the amount of released drug in a phagolysosomal-like milieu for correlating the results with the later performed Enzyme Linked Immunosorbent Assay (ELISA) for cytokine determination. For the procedure, the formulation was placed in different micro reaction tubes under sink conditions and incubated at 37 °C with gentle shaking for 1, 2, 3, 4, 6, 24 and 48 h. To analyse the released amount, the tubes were centrifuged at 24,000×*g* for 30 min. The resulting supernatant was then analysed using a microplate spectrophotometer. To determine the curcumin content, the supernatant was mixed 1:1 with ethanol to facilitate dissolution of the compound and measured at λ_ex_ = 430 nm and λ_em_ = 535 nm. For quantification of the released oligonucleotides, the supernatant was mixed with ethidium bromide followed by fluorescence measurement at λ_ex_ = 526 nm and λ_em_ = 605 nm [59]. Each value was determined as independent triplicate.

### 2.12. Quantification of TNF-α Release by Enzyme Linked Immunosorbent Assay (ELISA)

To detect the influence of the developed formulations on the cytokine release (TNF-α) of the macrophage-like dTHP-1 cells after induced inflammation by lipopolysaccharide (LPS), an ELISA was performed [60]. A total of 200,000 THP-1 cells were seeded per well in a 24-well plate and differentiated with PMA to M0 macrophages as described in the section above. Subsequently, the cells were incubated with 500 µL of the different microrod formulations and a concentration of 40, 200 and 400 µg/mL for 2 days at 37 °C in a humidified atmosphere of 5% carbon dioxide. These concentrations represent the same concentrations per cell as used in the cytotoxicity assay. After incubation, the supernatant was discarded and replaced by 500 µL of 200 µg/mL LPS from Salmonella typhimurium for 6 h to induce inflammation of the macrophages leading to an increased release of proinflammatory cytokines such as TNF-α [61]. The resulting supernatants were then centrifuged at 200× *g* for 10 min to remove cell residues and finally analysed by ELISA for the cytokines. As a positive control (PC), dTHP-1 stimulated with LPS for 6 h without treatment were used. Untreated cells served as a negative control (NC). Further controls included the plain drugs curcumin and siRNA without a carrier system, as well as scrambled siRNA with and without carrier were used to exclude possible unspecific reactions with oligonucleotides.

### 2.13. Scanning Electron Microscopy for Analysis of Rod Morphology

Morphology of the nanostructured microrods was determined by scanning electron microscopy (SEM). For visualization, a Zeiss Evo HD 15 Electron Microscope (Carl Zeiss AG, Jena, Germany) was used, equipped with a Lanthanum hexaboride (LaB_6_) cathode. To prepare the samples for the measurement, the microrods were redispersed in water and a few drops of the suspension were placed on a silica wafer and left to dry. These samples were then sputtered with an approximately 10 nm thick gold layer using a Quorum Q150R ES sputter coater (Quorum Technologies Ltd., East Grinstead, UK) to minimize surface artifacts and allow high magnifications [62]. The images were taken at a voltage of 5 kV and a magnification of 15 kX.

### 2.14. Confocal Laser Scanning Microscopy for Particle Visualization

In addition to SEM visualization, confocal laser scanning microscopy (CLSM) (LSM710, Carl Zeiss AG, Jena, Germany) was used to characterize the morphology of the microrods. Therefore, the formulation was redispersed in water and dropped onto a glass slide and covered with a cover glass. To visualize the rhodamine B-labelled microrods, λ_ex_ = 545 nm and λ_em_ = 567 nm were used; for curcumin, λ_ex_ = 440 nm and λ_em_ = 542 nm; and for the siRNA stained with ethidium bromide, λ_ex_ = 526 nm and λ_em_ = 605 nm. For the measurement, a water corrected objective M27 was used with a magnification of 40× and a numerical aperture of 1.2.

### 2.15. Statistical Evaluation

Two-sided Student’s *t*-test from Excel 365 software was used to check for significance between two measurements. A significant change was adopted for *p* < 0.05.

## 3. Results

### 3.1. Particle Characterization

In a first step, the manufactured nanostructured cylindrical microparticles (microrods) consisting of 260 nm mesoporous silica nanoparticles (mSNPs) were characterized by observing the morphology using SEM and CLSM. Therefore, unlabelled nanoparticles were stained with rhodamine B to generate a fluorescence signal for CLSM imaging on the one hand and to be able to perform quantification by fluorometry in the NGI experiments on the other hand. Due to the large number of hydroxy groups on the surface of the mSNPs and the resulting strongly negative zeta potential of −40.03 ± 0.80 mV (Table 1), these groups are well suited for chemical modification of the surface [63]. Commonly, 3-aminopropyltrimethoxy silane (APTS) is used as a coupling agent preparing the surface of silica nanoparticle to be stained with dyes [35]. As demonstrated in Table 1, surface modification with APTS led to a change in zeta potential to 36.07 ± 0.57 mV, which turns less positively charged after coupling rhodamine B to the amino groups (22.70 ± 1.06 mV), but is still sufficient to be coated during the layer-by-layer procedure. The size of the particles remains constant due to the modification of the surface, with a small decrease after covalent binding of rhodamine B. This phenomenon can be explained by the fact that rhodamine B reduces the polarity of the particles, which results in reduced water association at the surface of the particles and therefore a decrease in the hydrodynamic diameter [64]. 

After staining, the mSNPs were first characterized by microplate spectrophotometer to check if the functionalization process was successful (Appendix A); afterwards, they were used as building blocks for the microrods. To characterize the mesoporous structure of the particles, they were imaged using TEM. Here, the porous structure and the associated large pore volume were clearly visible (Appendix A). In the beginning, an unloaded batch was produced to verify the process of staining and subsequent stabilization by the layer-by-layer technique to the microrods. As illustrated in Figure 1, the formation of such nanostructured cylinders could be successfully achieved as demonstrated by SEM (Figure 1a,b). The successful modification with the fluorescent probe is clearly visible by CLSM, showing the respective cylindrical form of the particles (Figure 1c). Both images depict cylinders in the expected size range. The template-assisted approach allowed the production of the microrods with a defined length of 10 µm and width of 3 µm. The microparticles seem to assemble in larger arrangements and piles, as obvious from the SEM micrographs due to drying. Looking at redispersed conditions in aqueous environment, it can be clearly seen that the particles are not agglomerated but well separated from each other. The Neubauer counting chamber was used to determine the number of particles per membrane, resulting in 6 × 10^6^ rods per membrane. The length-width ratio (aspect ratio) chosen here is 3.3, which is assumed to delay uptake of the particles into macrophages compared to spherical particles [65]. This prolongation of internalization favours longer retention in the lungs and slower degradation of the active ingredients, thus increasing effectiveness [66].

### 3.2. Evaluation of Aerodynamic Properties by Next Generation Impactor

After verifying the morphology of the fluorescently labelled microrods, the flight characteristics were determined by NGI. The determination of these parameters is of interest to get an estimation if the formulation may be able to reach the deep lungs. To enable deposition in the deep lungs, an aerodynamic diameter of 1–5 µm must be realized [67]. For the microrod formulation, an MMAD of 2.85 ± 0.44 µm was determined, which is favourable for deposition in the deep lungs [68]. The percentage of the mass of particles < 5 µm, represented by the FPF [69], is with a value of 32.86 ± 2.93% in an acceptable range compared to many commercially applied DPIs [70]. This value is mainly promoted by coating with L-leucine, which reduces hygroscopicity and increases surface roughness, resulting in a powder with less agglomeration and better flow properties [48,49,50]. 

Having a closer look at the MMAD of the microrods, it is evident that the value of 2.85 µm is lower than both the length and width of the particles. This is in accordance with Equation (2), correlating the aerodynamic diameter with the volume equivalent sphere [71,72].
(2)$$d_{a}={\;d}_{\mathrm{ve}\;}\sqrt{\frac{1}{\chi }\;\frac{\rho_{p}}{\rho_{0}}\;\frac{C_{c}\left(d_{\mathrm{ve}}\right)}{C_{C}\;\left(d_{a}\right)}}$$d_a_ = aerodynamic diameter;d_ve_ = volume equivalent diameter;χ = dynamic shape factor;$\rho_{p}$ = particle density; $\rho_{0}$ = standard density;C_c_(d_ve_) = Cunningham slip correction factor of the volume equivalent diameter;C_c_(d_a_) = Cunningham slip correction factor of the aerodynamic diameter.


The important parameter in this equation for aspherical particles is the dynamic shape factor χ. Calculating this factor according to Sturm [73] results in a value of 1.47 for an aspect ratio of 3.3. This means that in comparison to spherical particles, where χ is equal to one [71], while keeping the other parameters of the equation constant, the value under the root decreases, leading to a lower aerodynamic diameter (d_a_) for the same volume equivalent diameter (d_ve_). This value underlines the advantageous properties of the nanostructured cylinders for aerosol application. Looking at the aerodynamic parameters with an MMAD of 2.85 ± 0.44 µm and an FPF of >30%, they are found in the relevant range for deep lung deposition, thus enabling lung application [67,74,75].

### 3.3. Drug Loading and Release Kinetics

The microrods were loaded with two different drugs exerting anti-inflammatory effects to generate a combined and enhanced biological effect in comparison to one drug alone, and thus the inhibition of TNF-α release after inflammation was stronger. Therefore, curcumin was loaded into the pores of the mSNPs, as they create a protective environment [18] which leads to a slower degradation of this instable drug [76]. To determine the highest possible loading of the pores, a BET measurement was performed. A pore volume of 0.1675 ± 0.0089 cm^3^/g was found, leading to a possible loading of approx. 22% loaded mass of curcumin with respect to the particles’ weight. To confirm this theoretical maximum loading experimentally, the mSNPs were loaded with 5–20% curcumin and then the pore volume was determined (Appendix A). The results showed that at 20% loading, only 0.0298 (cm^3^/g) of the pore volume remained free, confirming the calculated maximum loading of approx. 22%. For the preparation, the amounts were chosen in such a way that a load of 20% (m/m) would be theoretically achievable. To show that only the pores are filled in the loading method performed, non-porous particles were also attempted to be loaded. No fluorescence signal was detected here, which excluded a loading of the surface with curcumin.

The siRNA was adsorbed to the surface of the microrods during the stabilization step using the layer-by-layer technique. In a first step, the loaded formulation was visualized with both SEM and CLSM, as the curcumin is fluorescent, and the siRNA used was stained with ethidium bromide. As shown in Figure 2, curcumin (C) and siRNA against TNF-α (D) can be visualized fluorometrically. The signal obtained indicates that the entire formulation is homogeneously loaded with both substances (inserts). When looking at the morphology under the SEM (Figure 2A,B), the nanostructured cylindrical shape of the microrods can be seen without affecting the stability. 

After successful loading, the release kinetics of the drugs were investigated. Therefore, an artificial phagolysosomal fluid (PSF) was prepared [58]. This medium was chosen for the in vitro release as the microrods are phagocytosed by macrophages due to their size [77] after reaching the lungs. Looking at the release kinetics in Figure 3, the siRNA shows a sustained release whereas the curcumin exhibits a delayed release. These release profiles correspond to the expected results, as the siRNA is directly available due to its localization on the surface of the microrods and thus offers a large area to be detached, whereas the curcumin is protected twice from release, once by the hydrophobic environment in the pores and secondly by the polymer layers covering the mSNPs. Consequently, the amount of curcumin released was below the detection limit during the first 24 h and after 48 h at 15 µg/mg microrods, which is about 8% of the total dose. For siRNA, approx. 540 ng/mg microrods were released, which is about 15% of the total amount. Thus, after 48 h, the siRNA was released roughly twice as fast as curcumin. The release profile of siRNA, which is characterized by a step pattern consisting of a sharp increase in release leading to a plateau followed by a slower increase, is favoured by several factors. These are, on the one hand, the swelling of the polymer layers, as well as an ion exchange with the surrounding counterions and the detachment of the various layers. This influences the release of the different polymers and thus also of the siRNA, which results in the release pattern observed [78,79].

### 3.4. Cell Viability of Alveolar Basal Epithelial Cells and Macrophages

After the particles were successfully loaded with the selected anti-inflammatory drugs, acute cytotoxicity was evaluated. Since during pulmonary application immunologically active cells as well as epithelial cells encounter the applied formulation, a toxicity test was performed on both A549 cells as model for alveolar epithelial cells as well as on dTHP-1 cells as a macrophage cell line. As illustrated in Figure 4, the MTT assay showed no toxicity for 10 to 100 µg/mL up to an incubation time of 48 h. Here, the loaded microrods represent 20% curcumin loading and siRNA against TNF-α in the outermost layer and the unloaded microrods are the formulation without any drug. The concentrations used per cell were chosen to provide the same conditions as in the ELISA performed subsequently. Since no toxicity was generated by the selected parameters, the formulation was used to determine the effect on the cytokine release.

### 3.5. TNF-α Quantification via ELISA

The successfully prepared nanostructured microrods loaded with curcumin and siRNA showing no toxicity on relevant time scales allowed the investigation of the cytokine inhibition potential of the microrods. The dTHP-1 cells were grown in 24-well plates with 2 × 10^5^ cells per well and incubated with 500 µL of 400 to 40 µg/mL microrods suspension for 48 h to generate the same rod-to-cell ratio as in the previous MTT assay (no toxicity observed). Subsequently, the cells were incubated for 6 h with LPS to induce an inflammation and afterwards cytokine release was measured. In addition to the formulation loaded with curcumin and siRNA, further control groups were tested for TNF-α inhibition allowing to separate the effect of the different compounds. Therefore, microrods only loaded with curcumin as well as curcumin solution were tested. To exclude unspecific gene silencing by an oligonucleotide sequence, a scrambled siRNA with the same number of base pairs was used as control. Additionally, all drugs were used without a carrier system to check their behaviour with respect to cytokine inhibition. As positive control (PC), dTHP-1 were cultivated without LPS stimulation whereas for the negative control (NC), cells were only incubated with LPS for 6 h without any drug delivery system or drug applied. Comparing the results of the ELISA (Figure 5), the microrods loaded with both anti-inflammatory reagents were found to be most powerful and inhibit the TNF-α release significantly in comparison to the NC, as well as the loaded formulation with just one drug. In addition, the unspecific siRNA and the pure drugs without a carrier system had no influence on cytokine inhibition, as expected. This demonstrates that loading the microrods with two drugs has a positive effect on TNF-α reduction. Furthermore, this approach is significantly more effective than applying just one drug. Additionally, the results obtained show that the developed nanostructured cylindrical microparticles loaded with curcumin and siRNA is a drug delivery system with suitable properties to reduce inflammation reactions of macrophages in the lungs. The release kinetics only providing a small fraction of the loaded drug released after two days might be also a beneficial aspect for future in vivo studies, as the reduction in the inflammation will not only last 48 h.

## 4. Discussion

This manuscript described the stepwise production of a nanostructured cylindrical microparticle system for pulmonary application, loading two anti-inflammatory drugs to reduce TNF-α release by human macrophage cell line THP-1. Mesoporous silica nanoparticles were used as building blocks of these microparticles as they have various positive properties in terms of drug delivery. Due to their large specific surface area and pore volume, high quantities of drugs can be loaded into the pores and additionally this porous structure leads to an improved biocompatibility [80]. Another advantage of pore loading is the protection of unstable drugs such as curcumin due to the protective environment created in the pores [18]. This property was taken advantage of when loading curcumin, as this anti-inflammatory drug [81] is subject to rapid degradation [29] and thus should be transported as closely as possible to the target. After loading the mSNPs, they had to be moved to a next larger dimension, since the aerodynamic properties were not sufficient to reach the deep lungs [67]. To allow for formation of inhalable objects, the layer-by-layer technique was used to increase the adhesion forces of the nanoparticles to each other and thus the integrity of the cylinders. Thus, a stable microparticulate formulation was generated. Various polyionic polymers can be used in this technique, but also biomolecules such as proteins, polypeptides and nucleic acids [82]. The aerodynamic properties of such a hierarchical system loaded with two types of anti-inflammatory drugs should offer a potential for lung deposition with values below the diameter of the template and the microcylinders (<3 µm). In the developed formulation, a specific siRNA against TNF-α was used as layering material to stabilize the formulation and to act at the same time as anti-inflammatory active ingredient. The delivery of siRNA faces many hurdles due to the chemical structure. The typically low stability, as well as the membrane impermeability caused by the polyanionic character are the biggest challenges [83]. As the target location of siRNA is the cytoplasm [84], branched polyethyleneimine was used as polycationic polymer for stabilisation, as this molecule has a strong buffering capacity, which promotes osmotic swelling of the lysosome and thus destabilizes the membrane of the vesicle, leading to diffusion of the siRNA in the cytoplasm (proton sponge effect) [43]. A provision of siRNA and curcumin without the delivery system did not impact the TNF-a secretion. However, the combination of both anti-inflammatory drugs with the developed drug delivery system showed a significant reduction in cytokine release compared to the individual agents. This suggests that the combination of two such drugs will allow stronger effects in reducing the inflammatory status.

The use of different particle shapes for drug delivery has been of increasing interest in recent years [85]. For lung application, a cylindrical shape is of interest, since on the one hand the aerodynamic properties are suitable for inhalation (as described in the manuscript), and on the other hand the particle–cell interaction is different to spherical particles. By changing the aspect ratio of the microrods, particle uptake into macrophages can be prolonged [33], which allows a sustained residence time of the formulation and the active ingredient. A sustained release was also previously demonstrated in vivo for cylindrical microparticles using plasmid DNA, where a biological output was detected after five days [25]. In addition, the surface of the cylindrical nanostructured microparticles is significantly higher than their spherical counterparts due to the large internal surface. This allows a larger amount of drugs to be loaded. Thus, larger quantities can be transported to the target cell population, which is particularly important for siRNA delivery [86]. Especially, the effective transport of siRNA into the lungs is still a major hurdle at present [87]. However, the data discussed above in terms of pulmonary deposition (NGI data) and the inhibition of TNF-a release (ELISA data) depict an intriguing direction with respect to co-application as well as with respect to novel formulation approaches.

## 5. Conclusions

The delivery of curcumin and siRNA is difficult due to chemical properties of the two compounds. Therefore, it is important to develop suitable drug carrier systems to enable effective transport. Due to the rapid decay of curcumin, protection of this substance is particularly important. This was provided by the pores of the mSNPs and the protective environment formed there, as well as the polymer layers above. The loading, as well as the transport of siRNA to macrophages, could be successfully evaluated, resulting in a significant reduction in TNF-α release. The combination of both anti-inflammatory drugs proved to be the most efficient. Besides the positive biological results, the obtained aerodynamic characteristics with a MMAD of 2.85 µm and a FPF > 30% render a possible pulmonary application feasible using such systems.

In summary, the developed nanostructured cylindrical particle formulation is suitable for the transport of labile drugs and, due to its flight properties, can enable transport into the lungs. In addition, the formulation is non-toxic to both alveolar epithelial cells and macrophages and can significantly inhibit TNF-α release after inflammation with LPS, making it a potential alternative to a drug delivery system for the lungs.

This approach can be used to deliver macromolecular and small drugs to efficiently treat pulmonary diseases. The specific interaction due to their size with macrophages in combination with the straightforward loading approach by adsorbing the macromolecular drug promise to be an interesting carrier system also for other diseases where macrophages play central roles in inflammation (e.g., COVID-19) or even bacterial infections such as tuberculosis. The variability of loading should allow a broad field of application; for this, the release must be controlled and further developed and the production must be further industrialized. With these issues addressed, the potential modulation of the biological effects, uptake time und connected residence time are promising targets.

## Figures and Tables

**Figure 1 pharmaceutics-13-00844-f001:**
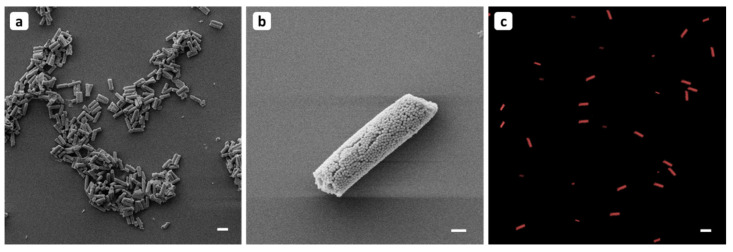
The microparticle formulation composed of mSNP functionalized with rhodamine B visualized (**a**,**b**) by SEM and (**c**) by CLSM. In all three images, the cylindrical shape can be seen. In (**a**), the particles are not well dispersed due to drying effects. With higher magnification used in (**b**), the nanostructure of the formulation is visible. (**c**) The formulation was visualised by CLSM with λ_ex_ = 545 nm, λ_em_ = 567 nm, dispersed in water. The homogeneous particle dispersion is clearly visible, and the fluorescent signal indicates that the functionalization with rhodamine B was successful. Representative sections were used for all images. Scale represents 10 µm for (**a**,**c**) and 1 µm for (**b**). The brightness of the CLSM image (**c**) has been increased for better visibility.

**Figure 2 pharmaceutics-13-00844-f002:**
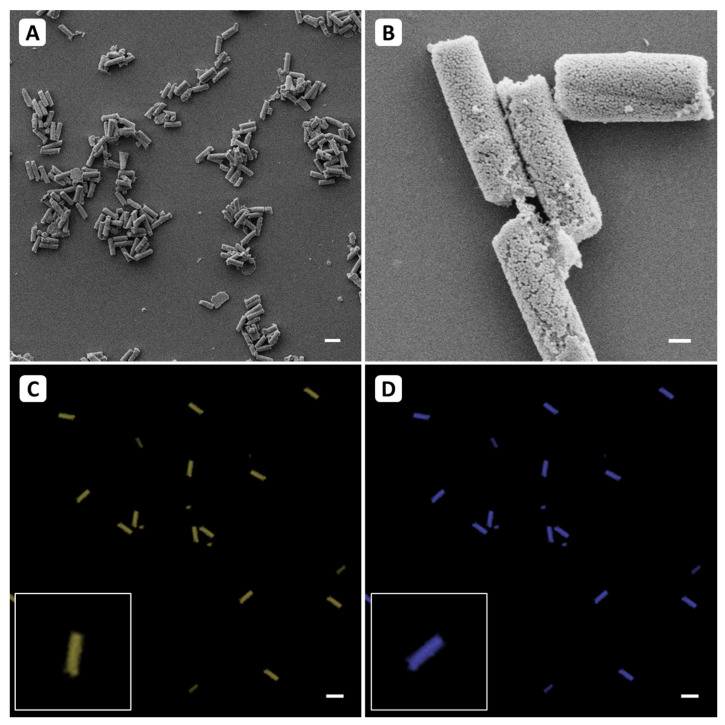
The curcumin and siRNA loaded formulation under the SEM (**A**,**B**) and CLSM (**C**,**D**). (**A**) The SEM image shows that neither the stability nor the morphology of the microrods are affected by drug loading. (**B**) Here, the nanostructure as well as the cylindrical shape is visible. (**C**) Visualization of curcumin fluorescence of λ_em_ = 542 nm was realized after excitation at λ_ex_ = 440 nm. Here, the homogeneous loading of the mSNPs can be seen. (**D**) To visualize the double-stranded siRNA used for CLSM measurement, the formulation was treated with ethidium bromide to generate intercalation, which allowed the detection of a signal at λ_ex_ = 526 nm, λ_em_ = 605 nm. The fluorescence is also homogeneously distributed over the whole microparticle surface, as shown in the magnified inset of (**C**,**D**), indicating a homogeneous loading along the particle. Representative sections were used for all images. Scale represents 10 µm for (**A**,**C**,**D**) and 1 µm for (**B**).

**Figure 3 pharmaceutics-13-00844-f003:**
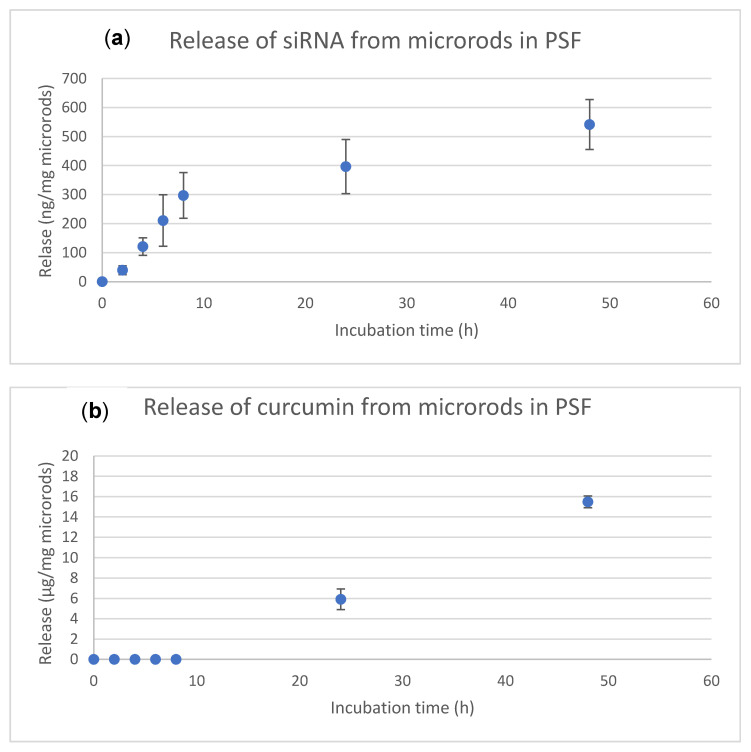
In vitro release kinetics of siRNA (**a**) and curcumin (**b**) in PSF for 48 h at 37 °C under gentle shaking. The release profiles show that the two drugs were released with different kinetics. The curcumin, which was located in the pores of the mSNPs and covered by various polymers, was only detected after 24 h. After two days, 15 µg per mg rods were released, which is about 8% of the total load. The siRNA found on the surface of the microrods was released continuously with a final amount of about 540 ng per mg microrods after 48 h, which is about 15% of the load. For each time point, three individual experiments were performed and the mean values as well as standard deviation were determined (n = 3).

**Figure 4 pharmaceutics-13-00844-f004:**
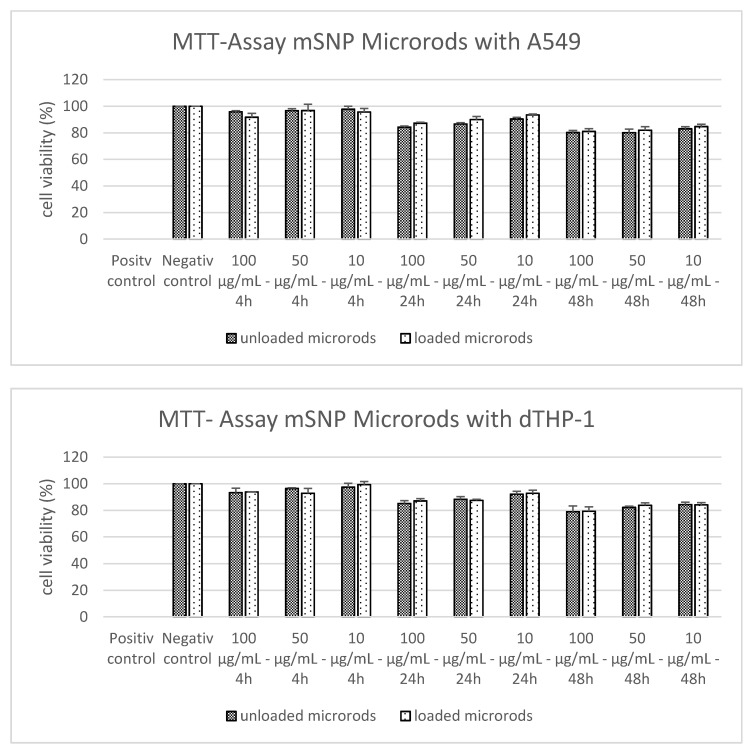
Determination of cell viability after incubation of the human alveolar epithelial cell line A549 (upper graph) and the differentiated human macrophage cell line dTHP-1 (lower graph) with unloaded and curcumin + siRNA loaded microrods. Incubation times of 4 to 48 h were chosen with a concentration of 10, 50 and 100 µg/mL. These parameters were adjusted to those of the ELISA performed later to see if the conditions are suitable for cytokine determination, as toxicity would cause high cytokine levels. Up to 48 h incubation time, no relevant toxicity can be observed.

**Figure 5 pharmaceutics-13-00844-f005:**
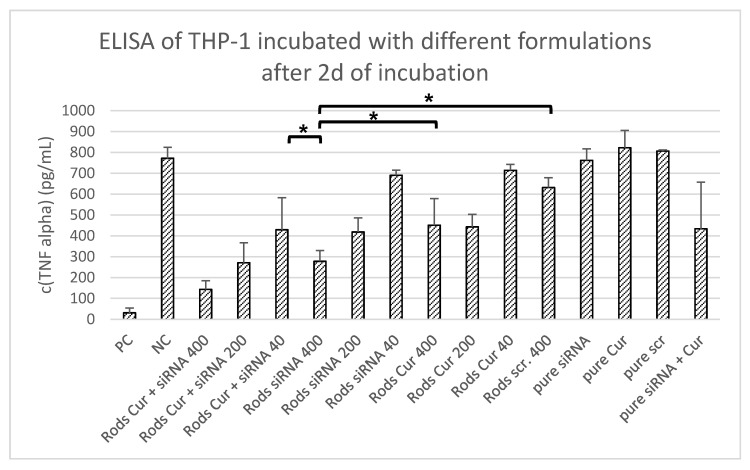
Determination of TNF-α release after incubation with different formulations for 48 h followed by LPS stimulation. As NC, cells were stimulated only with LPS without any carrier system, whereas the PC represents unstimulated cells. In order to compare the microrods loaded with both active ingredients (Rods Cur+siRNA) and the individually loaded microrods, a formulation containing only curcumin (Rods Cur) and only siRNA (Rods siRNA) were prepared and its influence on cytokine release determined. To exclude possible unspecific gene silencing by an oligonucleotide sequence, a scrambled siRNA with the same number of base pairs was used as an additional control. In addition, the drugs were applied to the cells without a carrier system to show the influence of the microrods as a means of transport. The results obtained show that the formulation with both active ingredients can significantly reduce the TNF-α release in contrast to the formulations with just one drug, or the active ingredients without carrier. Moreover, no unspecific inhibition by scrambled siRNA was observed. However, it can also be seen that all microrod formulations can inhibit cytokine release, which shows that the cylindrical microparticles can effectively transport the active ingredients into the macrophages. Significance of the obtained data was calculated by two-sided Student’s *t*-test. * *p* < 0.05.

**Table 1 pharmaceutics-13-00844-t001:** Hydrodynamic diameter, PDI and zeta potential of the unmodified mSNPs, covalently bound APTS and with rhodamine B and APTS-modified mSNPs. The numbers presented show that APTS modification leads to a change of zeta potential from negative to positive values without influencing the hydrodynamic diameter.

	Unmodified mSNPs	APTS Modified mSNPs	Rhodamine B + APTS Modified mSNPs
Hydr. Diameter (nm)	262.13 ± 3.98	263.23 ± 1.33	245.8 ± 2.97
PDI	0.102 ± 0.013	0.045 ± 0.022	0.084 ± 0.068
Zeta potential (mV)	−40.03 ± 0.80	36.07 ± 0.57	22.7 ± 1.06

## Data Availability

Not applicable.

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
