# Peer review of "Cylindrical Microparticles Composed of Mesoporous Silica Nanoparticles for the Targeted Delivery of a Small Molecule and a Macromolecular Drug to the Lungs: Exemplified with Curcumin and siRNA"

_pharmaceutics, 2021, doi:10.3390/pharmaceutics13060844_

Round 1

Reviewer 1 Report

The manuscript "Cylindrical microparticles composed of mesoporous silica nanoparticles for the targeted delivery of a small molecule and a macromolecular drug to the lungs: exemplified with curcumin and siRNA" by Thorben Fischer, Inga Winter, Robert Drumm, and Marc Schneider focus the use of cylindrical microparticles composed of mesoporous silica nanoparticles for the targeted delivery of a small molecule and a macromolecular drug to the lungs: exemplified with curcumin and siRNA. The subject is of actual interest in the drug delivery area and could catch researchers attention worldwide. The manuscripts has to be revised before publication. Some comments are following listed.

  • The authors should stress more on the novelty of the study, in the introduction. They should address the difference of their study in comparison to others.
  • Mesoporous silica has been used in another studies, which wre not cited in text, as: Yadav YC, Pattnaik S, Swain K. Curcumin loaded mesoporous silica nanoparticles: assessment of bioavailability and cardioprotective effect. Drug Dev Ind Pharm. 2019 Dec;45(12):1889-1895. doi: 10.1080/03639045.2019.1672717; Kong ZL, Kuo HP, Johnson A, Wu LC, Chang KLB. Curcumin-Loaded Mesoporous Silica Nanoparticles Markedly Enhanced Cytotoxicity in Hepatocellular Carcinoma Cells. Int J Mol Sci. 2019;20(12):2918. doi:10.3390/ijms20122918; Curcumin-loaded PEGylated mesoporous silica nanoparticles for effective photodynamic therapy, Gaizhen Kuang, Qingfei Zhang, Shasha He and Ying Liu, RSC Adv.,2020,10,24624–24630
  • Did the authors calculate a functionalization degree of the silica nanoparticles?
  • As well, what the loading capacity of the used silica nanoparticles?
  • The authors, affirmed that the "BET measurement to evaluate pore volume and loading with curcumin" has been performed, but in the manuscript body no information about could be found. Even in supplementary.
  • Did the authors register FTIR spectra on loaded/unloaded silica nanoparticles? How they prove the silica was functionalized/loaded with organic species?
  • How uniform/non-uniform are pores of the used silica?
  • Do the authors have any result on as-commercialized mSNP, non-cylindrical microparticles, regarding the curcumin and siRNA loading? What the difference is between as-commercialized mSNP and cylindrical microparticles?

Reviewer 2 Report

REVIEWER’S COMMENTS

The manuscript Cylindrical microparticles composed of mesoporous silica nanoparticles for the targeted delivery of a small molecule and a macromolecular drug to the lungs: exemplified with curcumin and siRNAby Fischer et al demonstrates that the developed nanostructured, cylindrical particle formulation is suitable for the transport of labile drugs into the lungs.

  1. What concentration/ amount of curcumin was loaded into the nanoparticles? Please provide this information in the methods section.
  2. Please provide the catalog numbers of all reagents/ kits/ materials used in this study.
  3. Methods section: line 267 – cells per well?
  4. Please add a brief paragraph on future directions at the end of the conclusions section.
  5. Please be consistent with the style of references. For example, reference #20.

Round 2

Reviewer 1 Report

The authors provided required supplementary information improving the manuscript. Now the manuscripts can be considered for publication in the Journal Pharmaceutics.